# Hypothermal opto-thermophoretic tweezers

Pavana Siddhartha Kollipara [1], Xiuying Li[2], Jingang Li[3,4], Zhihan Chen[3], Hongru Ding [1], Youngsun Kim [3], Suichu Huang[5], Zhenpeng Qin [2,6,7,8] & Yuebing Zheng [1,3] ✉

Optical tweezers have profound importance across fields ranging from manufacturing to biotechnology. However, the requirement of refractive index contrast and high laser power results in potential photon and thermal damage to the trapped objects, such as nanoparticles and biological cells. Optothermal tweezers have been developed to trap particles and biological cells via opto-thermophoresis with much lower laser powers. However, the intense laser heating and stringent requirement of the solution environment prevent their use for general biological applications. Here, we propose hypothermal opto-thermophoretic tweezers (HOTTs) to achieve low-power trapping of diverse colloids and biological cells in their native fluids. HOTTs exploit an environmental cooling strategy to simultaneously enhance the thermophoretic trapping force at sub-ambient temperatures and suppress the thermal damage to target objects. We further apply HOTTs to demonstrate the three-dimensional manipulation of functional plasmonic vesicles for controlled cargo delivery. With their noninvasiveness and versatile capabilities, HOTTs present a promising tool for fundamental studies and practical applications in materials science and biotechnology.

The development of optical tweezers has led to tremendous advances in many fields such as optical nanomanufacturing[1], microrobotics[2], cell mechanics[3], and nanomedicine[4]. Optical tweezers trap target objects by the gradient force, which depends on the refractive index, particle size, and laser wavelength[5]. High laser power is usually required to trap nanomaterials and biological objects that have a low refractive index contrast with their surroundings, which can induce damage to the materials and reduce the cell viability[6].

To overcome these challenges, different variations of optical tweezers, such as plasmonic tweezers[7–10], opto-electronic tweezers[11,12], and opto-acoustic tweezers[13], have been developed. However, they are usually limited by specific substrates, complex setups, or confined working ranges. Recently, optothermal tweezers have been developed to achieve versatile manipulation of colloidal particles under a light-controlled temperature gradient[14–18]. While optothermal tweezers enable enhanced trapping capability with a laser power that is 2–3 orders of magnitude lower than optical tweezers[19], optical heating can cause thermal stress and degradation to the particles and biological cells. In addition, since many colloids and cells show a thermophobic behavior and move away from the laser heating spot, optothermal tweezers require additional surfactants or salts to tune their thermophoretic response[20–23]. However, trapping cells and biological objects in required fluidic environments is often essential for biological applications to reduce the effect of additives and elucidate the bio-physio-chemical interactions of the cells[24–26]. Without any solution modification, specialized traps such as thermal Paul trap and anti-Brownian electrokinetic trap were implemented to trap

---

[1]Walker Department of Mechanical Engineering, The University of Texas at Austin, Austin, TX 78712, USA. [2]Department of Mechanical Engineering, The University of Texas at Dallas, Richardson, TX 75080, USA. [3]Materials Science and Engineering Program and Texas Materials Institute, The University of Texas at Austin, Austin, TX 78712, USA. [4]Laser Thermal Laboratory, Department of Mechanical Engineering, University of California, Berkeley, CA 94720, USA. [5]Key Laboratory of Micro-Systems and Micro-Structures Manufacturing of Ministry of Education and School of Mechatronics Engineering, Harbin Institute of Technology, Harbin 15001, China. [6]Department of Bioengineering, The University of Texas at Dallas, Richardson, TX 75080, USA. [7]Department of Biomedical Engineering, The University of Texas Southwestern Medical Center, Dallas, TX 75390, USA. [8]Center for Advanced Pain Studies, The University of Texas at Dallas, Richardson, TX 75080, USA. ✉e-mail: zheng@austin.utexas.edu

thermophobic objects with limited accuracy and manipulation abilities[27–29].

In this work, we propose hypothermal opto-thermophoretic tweezers (HOTTs) to overcome these limitations. Specifically, we couple environmental cooling and localized laser heating to achieve low power thermophoretic trapping of target objects and simultaneously avoid optical and thermal damage. More importantly, this cooling strategy also plays a vital role in facilitating the thermophilic behavior to enable the trapping of diverse colloids at different conditions. We also demonstrate the successful trapping and manipulation of fragile erythrocytes in different tonicities to resemble different bio-physio-chemical functionalities. We further show the capability of HOTTs for three-dimensional manipulation (3D) of plasmonic vesicles for light-controlled drug delivery.

## Results and discussion
### Working principle of HOTTs

Figure 1a, b depicts the general schemes of opto-thermophoretic trapping at ambient temperature and under environmental cooling, respectively. A thermoplasmonic substrate is used to generate a temperate gradient ($\nabla$T) under local laser heating (Supplementary Fig. 1, also see Methods). The particle under the temperature gradient is subject to a thermophoretic force ($\mathbf{F_{th}}$)[20,30], which can be expressed as

$$\mathbf{F_{th}} = -k_B T S_T \nabla T \qquad (1)$$

where $S_T$ is the Soret coefficient of the particle, $k_B$ is the Boltzmann constant, and T is the average temperature around the particle (see Supplementary Note 1). Here, thermophoretic force includes the contributions from all thermal-gradient-induced forces such as thermo-electricity, thermo-osmosis, and thermo-diffusion. The direction of thermophoretic force is dependent on the sign of $S_T$ and a positive (or negative) $S_T$ leads to a repulsive (or attractive) force. $S_T$ is a function of many parameters, including colloid composition, ionic concentrations, surface effects, particle size, and temperature[31,32]. Mainly, temperature plays a significant role in affecting the particle-solvent interactions. The aqueous solvent used in several biological applications exhibits thermo-polarization of water molecules, creating a thermoelectric field that inverts as the temperature changes. This is

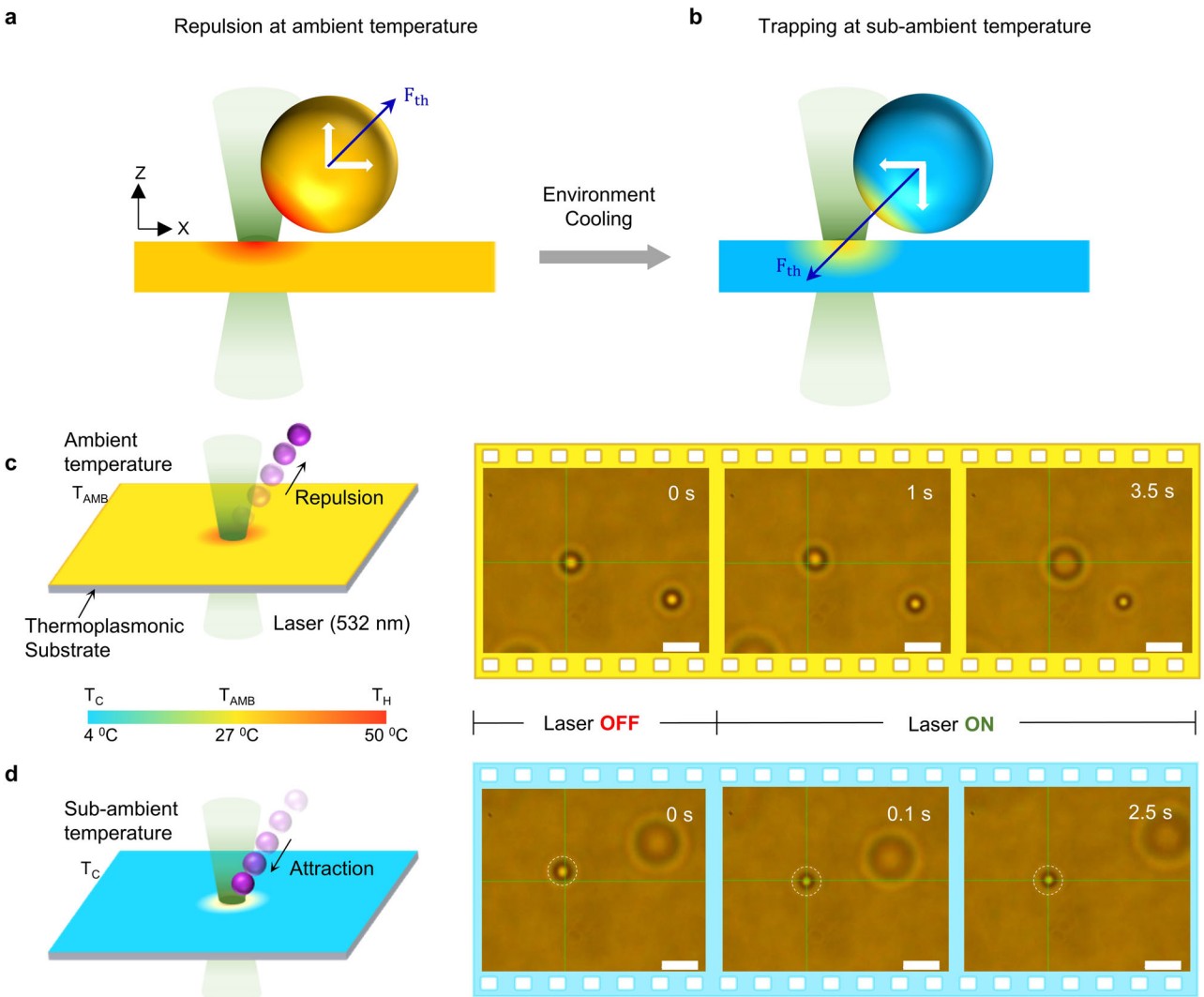

**Fig. 1 | Working principle of HOTTs.** (**a**) At ambient temperature, thermophoretic force ($\textit{F}_{th}$) repels the particle away from the laser in most conditions. White arrows indicate $\textit{F}_{th}$ decomposed along and perpendicular to the substrate (**b**) In HOTTs, $\textit{F}_{th}$ becomes attractive to trap particles at a sub-ambient temperature. **c** Schematic and timelapse optical images showing the repelling of a 1 μm PS particle in DI water by the laser beam at an ambient temperature of 27 °C. **d** The same particle was trapped at the laser beam at a sub-ambient temperature of 4 °C. The green crosshair indicates the laser beam center. Laser wavelength: 532 nm, laser power: 40 μW, beam radius: 850 nm, scale bars: 2 μm.

mainly due to the strong dependence of the quadrupolar component of the thermoelectric field on the temperature that is determined by the thermal expansion of water[33]. The resultant $S_T$ typically reduces with the decreasing temperature in a majority of aqueous solutions (Table S1), which can be described by an empirical equation[34]

$$S_T(T) = S_{T,\infty}\left(1 - e^{\frac{T^* - T}{T_0}}\right) \qquad (2)$$

where $S_{T,\infty}$ is the high-temperature limit, $T^*$ is the transition temperature where $S_T$ changes the sign, and $T_0$ represents the strength of the temperature effect. $T^*$ changes mainly with the particle material that affects the particle-solvent interactions. This is due to the reorganization of the hydrogen bond network of the solvent while accommodating the dispersed particle[35]. At the ambient temperature, $S_T$ is positive for most objects and the thermophoretic force repels them away from the laser (Fig. 1a). In HOTTs, we adopt an environment-cooling strategy to enable a negative $S_T$ and a thermophoretic attractive force to trap the particle at the hotspot (Fig. 1b). A custom temperature controller based on Peltier cooling is designed to enable fast cooling of the sample (Supplementary Fig. 2 and Supplementary Note 2). As a proof-of-concept demonstration, we compared the trapping behaviors of $1\,\mu m$ polystyrene (PS) microparticle in deionized (DI) water at two different conditions. At the ambient temperature of 27 °C, the PS microparticle was repelled away from the laser beam due to a net repulsive thermophoretic force (Fig. 1c). In contrast, when the environmental temperature was cooled down to 4 °C, the particle was successfully trapped at the laser beam center by the thermophoretic attraction force (Fig. 1d and Supplementary Movie 1).

## Versatility of HOTTs

Next, we demonstrate the use of HOTTs to trap diverse microparticles in different conditions to demonstrate its wide applicability (Fig. 2, Supplementary Fig. 3, and Supplementary Movie 2–4). In all cases, thermophoretic trapping of colloids (e.g., PS and silica microparticles) is enabled or significantly enhanced in HOTTs at a reduced ambient temperature. We measured the trapping stiffness to examine the trap strength at different temperatures (Supplementary Note 3). Figure 2a shows the trajectories of a $1\,\mu m$ PS particle trapped in $3\,mM$ $NaCl_{0.2}OH_{0.8}$ solution at varying temperatures. Here, we use an electrolyte solution dominated by NaOH to ensure trapping even at ambient temperature to determine the trend of the trapping strength over a higher range of temperatures. As the temperature reduces, the particle becomes more confined with respect to the laser beam center. Figure 2b further shows the calculated trapping stiffnesses of trapped particles at varying sample temperatures with different laser powers (0.05 mW, 0.14 mW, and 0.24 mW), sizes ($1\,\mu m$ and $9.5\,\mu m$), materials (PS and $SiO_2$), and solutions (DI water and electrolytes). In all the conditions, the trapping stiffness increases by 3–5 times with the reduced environmental temperature, showing the versatility of HOTTs. Moreover, by varying the composition of the electrolyte ('x' in $NaCl_xOH_{1-x}$), we demonstrate that trapping ability is induced in a wider range of compositions at reduced environmental temperature (Supplementary Fig. 4).

To quantitatively evaluate the effect of environmental temperature on the thermophoretic behavior, we measured the drag velocity of trapped $SiO_2$ particles in DI water to extract the Soret coefficient ($S_T$) at different temperatures (Supplementary Note 4). The experimental data fit nicely with the empirical formula (Eq. 2). It is noted that the Soret coefficient of the particle is tuned from $-0.1\times10^3\,K^{-1}$ to $-2\times10^3\,K^{-1}$ after reducing the temperature from 31 °C to 4 °C, which is of the same order of magnitude for PS particles of similar size[36]. The direction of thermophoretic force inverts

and causes attraction, and the magnitude of the force increases more than 10-fold, which contributed to the increased trapping stiffness.

The thermophoretic response of the particles also depends on the colloidal concentration[37,38], an important parameter in drug delivery and therapeutics applications. Stable trapping at any required concentration is considered essential for bio-statistical analysis and cell-cell interactions. However, the non-linear nature of the thermophoretic response of the particles due to collective effects and spatially varying temperature gradient makes it challenging to achieve trapping at all concentrations. Here, we show that HOTTs enable the consistent trapping of colloids at concentrations spanning across several orders. As a case study, $1\,\mu m$ PS particles are trapped using HOTTs at varying particle concentrations and sample temperatures. The trapping performance is evaluated with the trapping probability calculated as $100 \times n_{trap}/(n_{trap} + n_{repel})$, where $n_{trap}$ and $n_{repel}$ are the number of particles that are trapped and repelled by the laser beam at the given conditions, respectively. At high colloid concentrations, the thermoelectric field due to the charged particles is locally altered by the inter-particle interactions. In one-dimensional temperature gradients generated by a typical thermophoresis setup, the colloid's concentration and zeta potential are sufficient to determine the thermophoretic response of the particle[39]. However, in temperature gradients spatially varying over the same length scale as the particle's size (~1 μm), the thermoelectric response is additionally dependent on the surrounding particle's position distribution, which continuously varies due to the Brownian motion of the particle, thus requiring statistical analysis to determine the thermoelectric response of single particles. Accordingly, we focus the laser beam on single particles to determine the instantaneous trapping probability, which fluctuates between 10–40% at the ambient temperature at concentrations from $\Phi > 0.01$, ($\Phi$ is the volume fraction of colloids, Fig. 2e). As the temperature decreases to 4 °C, the enhanced thermophilic nature of the particles increases the trapping probability to 100%. At low colloid concentration ($\Phi < 0.01$) in the single-particle limit, the PS particle undergoes a transition from repelling at ambient temperature to trapping at low temperatures (Fig. 2d). Although single-particle trapping (or repulsion) is achieved without other particles in the trapping zone, the collective effects due to inter-particle interactions still exist at volume fractions as low as ~0.001[39]. At lower concentrations, a delicate balance between the collective effects and individual particle's response occurs, and an inflection point is typically observed with changing colloidal concentration[40], resulting in the least trapping performance at 2% at ambient temperature. However, when the temperature is reduced to 4 °C, a trapping probability of 100% is observed for all concentrations (Fig. 2e), highlighting the potential applications of HOTTs for trapping in complex fluids and highly scattering media, in vitro drug efficacy, and crystallization studies.

## Trapping of erythrocytes in distinct tonicities using HOTTs

Erythrocytes (also known as erythrocyte cells or red blood cells) are important biological entities that are currently used in drug delivery[41,42] and disease diagnostics[43,44]. Optical trapping of erythrocytes has promoted the understanding of cell mechanics and cell-cell interactions[45]. However, these studies are based on the local photo-deformation of the cell membrane and are limited to mature erythrocytes in isotonic solution only. Erythrocytes in different tonicities (hypertonic and hypotonic) serve as potential markers for pathophysiological disease diagnostics like sickle cell anemia[46] and malaria[47]. The tonicity of extracellular fluid alters the shape and size of erythrocytes, which has been recently used to determine the severity of illness caused in SARS-COV-2 patients[48]. Although optical trapping of erythrocytes in varying tonicities can enable the extension of these cell-cell interactions and biomedical studies for diverse diseases, it is

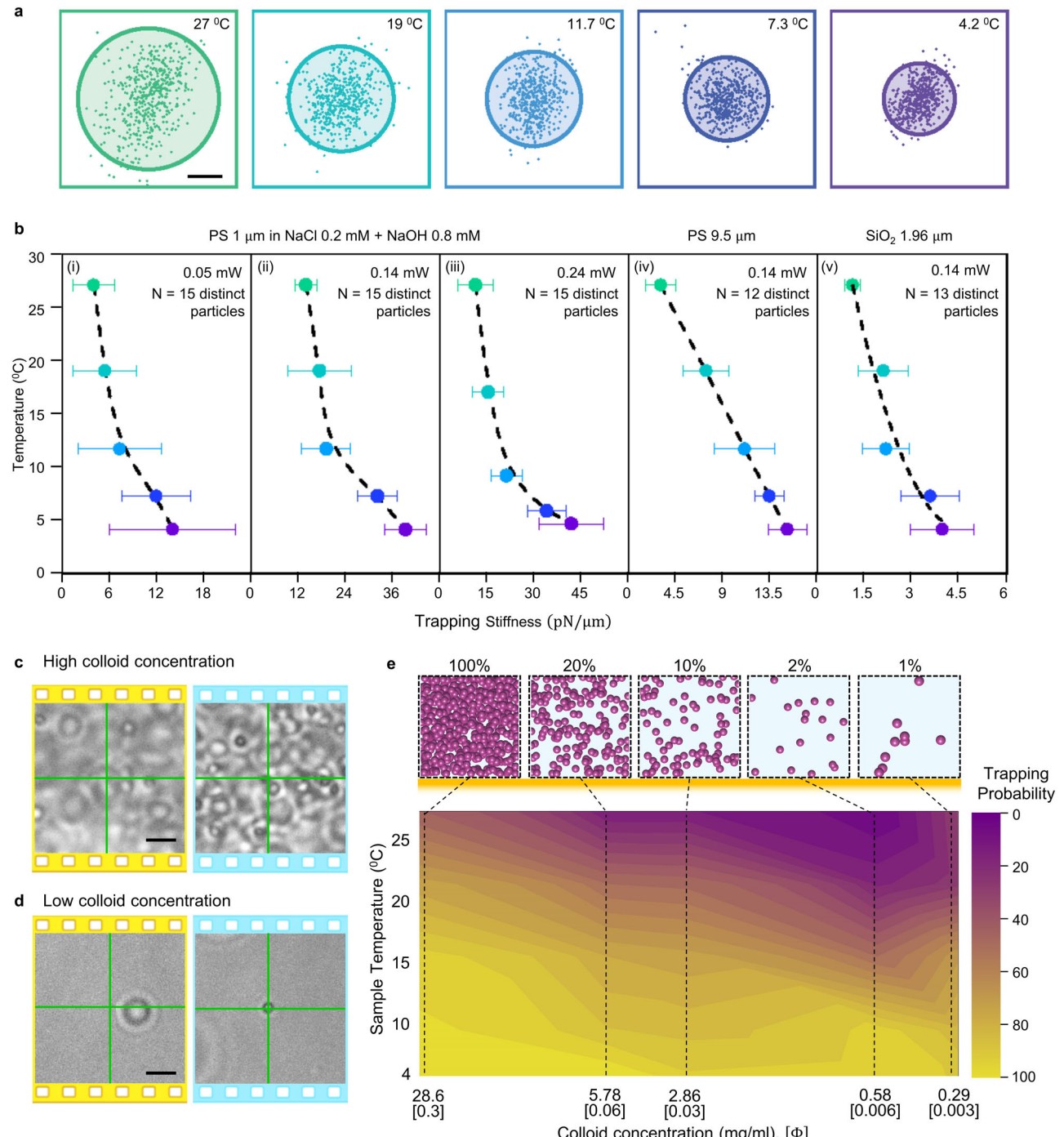

**Fig. 2 | Performance evaluation of trapping microparticles using HOTTs.**
(**a**) Particle trajectories of 1 μm PS particles (low colloid concentration) in 1 mM
$NaCl_{0.2}OH_{0.8}$ solution at varying environmental temperature. **b** Trapping stiffness
dependence on sample temperature at single-particle concentration indicates the
enhancement of trapping efficiency at lower temperatures for varying laser powers
(i–iii), sizes (ii, iv), materials (ii, v), and solutions (i–iii, iv-v). Data is presented as
mean values ± standard error of mean. **c**, **d** Optical images of repulsion at 27 °C
(yellow panels) and trapping at 4 °C (blue panels) of 1 μm PS particle in DI water at a
high concentration of 28.6 mg/mL (**c**) and low concentration (**d**) of 0.29 mg/mL.
Laser power is 50 μW. **e** Trapping probability of 1 μm PS particles as a function of
sample temperature and colloidal concentration. The highest concentration and
volume fraction (Φ) of 1 μm PS particles at 100% relative concentration is 28.6 mg/
mL and 0.3, respectively. Laser power is 50 μW. Scale bars: (**a**) 30 nm (**c**, **d**) 5 μm.

challenging because of the erythrocyte's non-linear optical
response[49,50] and photo-damage[51].

Here, we demonstrate the capability of HOTTs for biological
applications by trapping erythrocytes in different tonicities at low laser
power, while retaining their structural integrity. The tonicity of the
extracellular fluid is altered by tuning the concentration of phosphate
buffered saline (PBS) solution (Supplementary Fig. 5). First, we

dispersed healthy human blood cells in an isotonic solution, where the
salinity of surrounding fluids matches that of the cells (see Methods).
The cells in such an isotonic environment are disc-shaped, as shown in
Fig. 3a. At ambient temperature, although cells can be trapped by the
laser beam, the slightly elevated temperature at the laser spot results in
cell lysis. After the cell rupture, the extracellular fluid fills up the cell,
and the lack of a refractive index contrast between the cell and the

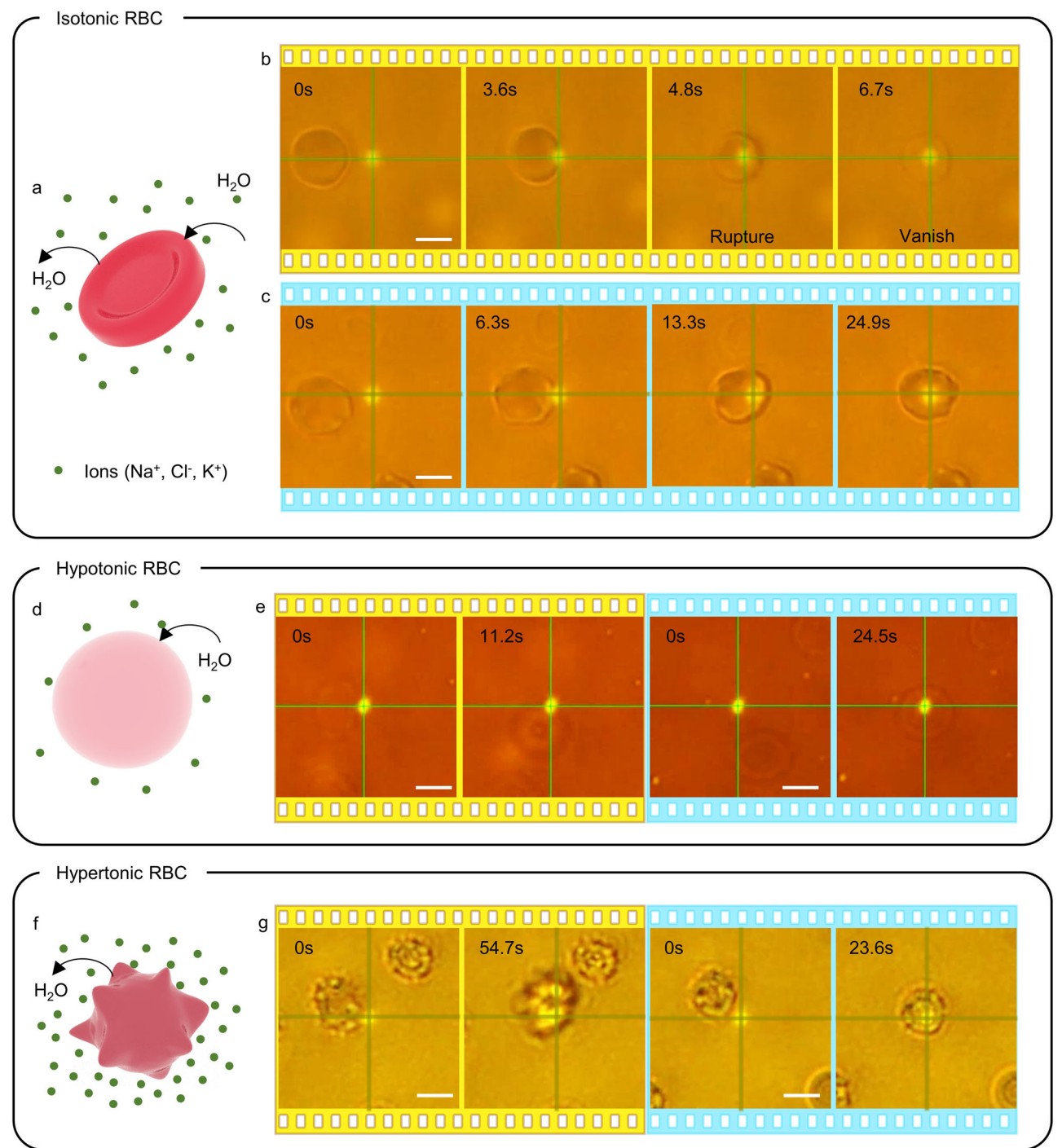

**Fig. 3 | Trapping erythrocytes in different tonicities using HOTTs.** (**a**) Schematic of erythrocytes in isotonic PBS. **b** Timelapse optical images showing the trapping and thermal rupture of erythrocytes at ambient temperature. Laser power: 0.67 mW. **c** Timelapse images of erythrocyte trapping at 4 °C at the same laser power. No cell rupture is observed. **d** Schematic of erythrocytes in hypotonic PBS. **e** Timelapse images of hypotonic erythrocytes repelled at ambient temperature (yellow panels) and trapped at 4 °C (blue panels). Laser power: 0.44 mW. **f** Schematic of erythrocytes in hypertonic PBS. **g** Timelapse images of hypertonic erythrocytes repelled at ambient temperature (yellow panels) and trapped at 4 °C (blue panels). Laser power: 0.34 mW. Scale bars: 5 μm.

surrounding solution results in the instantaneous vanishing of the erythrocyte under the microscope (Fig. 3b). To avoid the lysis, we reduce the sample temperature to 4 °C to stably trap the cell by HOTTs and simultaneously retain the integrity of the cellular membrane (Fig. 3c, Supplementary Movie 5).

When erythrocytes are dispersed in a low tonicity solution (hypotonic), the salt concentration in the extracellular fluid is lower than the erythrocyte. The osmotic gradient across the membrane results in the permeation of water into the cell and the swelling of the cell to become nearly spherical (Fig. 3d). The cell is repelled by the laser at ambient temperature and trapped by the enhanced thermophoretic force at 4 °C (Fig. 3e and Supplementary Movie 6). Last, the high salt concentration of the extracellular fluid shrivels the erythrocyte in high tonicity solution (hypertonic) (Fig. 3f). Like the hypotonic case, cells can only be trapped by HOTTs at the sub-ambient temperature (Fig. 3g and Supplementary Movie 7). Altering the tonicity

of the extracellular fluid changes the thermoelectric and thermo-osmotic fields, as well as the effective shape and size of the cells, which changes the Soret coefficient of the cells[20,52]. Despite the variation, erythrocytes are trapped by HOTTs in all tonicities, while maintaining their structural integrity.

While their structural integrity remains intact, further assay experiments are necessary to verify the preservation of cellular physiological conditions following laser exposure. Decreasing the duration of laser exposure would aid in maintaining the physiological conditions of the cells. Incorporating alternative engineering methods that employ focused laser beams to attract cells while gradually increasing the focal beam waist as cell approaches the laser beam (thus reducing the temperature gradient) would additionally contribute to the preservation of cellular integrity.

### 3D manipulation of plasmonic vesicles and controlled cargo release

Extracellular and synthetic vesicles have shown great importance in bioimaging, drug delivery, biological transport processes, and therapeutics[53–55]. Plasmonic vesicles are gold-coated vesicles with controlled optical and spectroscopic properties for diverse biomedical applications[56–59]. Although optical and optothermal trapping of naked vesicles or gold nanoparticles has been achieved[60,61], optical trapping of plasmonic vesicles is challenging due to the large plasmon-enhanced scattering force because of the gold layer. Also, the heat produced during laser illumination causes an uncontrollable thermophoretic force which usually directs the vesicle towards the cold (away from the laser beam) at ambient temperature[62]. In addition, the heat generated also induces drug release from plasmonic vesicles. The capability of trapping while maintaining the integrity of plasmonic vesicles will enable precise positioning followed by optically triggered drug release, and it holds great promises in several applications.

Here, we demonstrate the trapping and 3D manipulation of plasmonic vesicles by HOTTs, followed by controlled cargo release using a dual laser beam setup. A 660 nm laser beam is utilized to manipulate the vesicle, while a 532 nm laser beam is utilized to rupture the vesicle. Under 660 nm laser irradiation, the gold coating on the plasmonic vesicles absorbs light to generate a highly localized temperature gradient across the vesicle (Fig. 4a), which creates a self-induced thermophoretic force on the vesicle[63]. Meanwhile, the high optical scattering force causes the vesicle to repel away from the focus of the laser beam at the ambient temperature irrespective of the thermophoretic force direction (Fig. 4b and Supplementary Movie 8). However, the trapping of plasmonic vesicles can be achieved by HOTTs at a sub-ambient temperature of 4 °C. In this case, the self-induced thermophoretic force becomes attractive and is greatly enhanced to overcome the optical repulsion force and enable the 3D trapping of the vesicle near the focal plane (Fig. 4b). After positioning the vesicle, the subsequent illumination of a 532 nm laser beam can generate intense heat to rupture the plasmonic vesicle and release the cargo (Fig. 4c).

We first showed the manipulation of a plasmonic vesicle in the vertical direction by simply tuning the focus position of the laser beam (Fig. 4d). The vesicle is steadily trapped and elevated for over 55 μm. Next, the vesicle is transported in the lateral plane to demonstrate the in-plane manipulation (Fig. 4e). We further demonstrated versatile 3D manipulation of plasmonic vesicles by HOTTs in highly complex and challenging environments (Supplementary Fig. 6 and Supplementary Movie 9). After the vesicle is transported to the target position, a 532 nm laser beam is excited to rupture the membrane to release the cargo (Calcein dye) as shown in Fig. 4f (also see Supplementary Fig. 7 and Supplementary Movie 10). We observed an increase in fluorescent intensity since the calcein dye is self-quenched in the plasmonic vesicle and emits a stronger fluorescence upon release and dilution. The manipulation and rupture of vesicles depend on the interfacial properties between the gold nanoaggregates and solvent, which is

independent of the cargoes within the vesicles. Therefore, HOTTs can be extended to deliver diverse cargoes using plasmonic vesicles as the delivery agent.

We have developed HOTTs for the on-demand manipulation of diverse microparticles and biological cells by exploiting the enhanced thermophilic nature at a sub-ambient temperature. The hypothermal environmental temperature further facilitates the noninvasive trapping of fragile objects (e.g., erythrocytes) by suppressing thermal damage. This capability is desired for a variety of biological applications, including drug development and cell-cell interactions. Although we successfully demonstrate the trapping of red blood cells in diverse tonicities while retaining the structural integrity, additional studies are essential to successfully characterize the metabolic activity of the trapped cells after laser exposure. We further exploited HOTTs for 3D manipulation of thermosensitive plasmonic vesicles and demonstrated the light-controlled cargo release. HOTTs can additionally be extended to non-plasmonic targets by using a plasmonic or a light-absorbing particle as a delivery agent[63]. With their versatility and general applicability, HOTTs will have many potential applications in disease diagnostics, thermal therapy, drug delivery, and microrobotic surgery.

## Methods
### Materials
PS microparticle suspensions of 0.96 μm and 2 μm are purchased from Thermo Fisher Scientific. 0.5 μm PS particles and 1.96 μm SiO$_2$ particles are purchased from Bangs Laboratories. COOH- functional PS particles of 2.67 μm particles are purchased from Bangs Laboratories. Erythrocytes are purchased from Human Cells Bio and stored at 4 °C. All experiments involving erythrocytes are performed within 4 weeks of the extraction date, and new erythrocyte samples were prepared for every experiment.

### Fabrication of thermoplasmonic substrate
Glass coverslips were triple-rinsed with iso-propyl alcohol and water and cleaned under a nitrogen gun. The coverslips were then loaded into a thermal evaporator (Kurt J Lesker Nano36), and 4.5 nm gold films are thermally deposited at a pressure of $1 \times 10^{-7}$ torr at a rate of 0.1 nm/s. Later, gold-deposited coverslips are thermally annealed at 550 °C for 2 h (ramp for 2 h, constant temperature of 550 °C for 2 h, and ramp for 2 h). For microparticle experiments, thermally annealed substrates are used as prepared after cleaning using DI water and nitrogen gun. For erythrocyte experiments, the substrates are immersed in a 1 mM 11-mercaptohexanoic acid in ethanol to prevent the adhesion of red blood cells onto the substrate. The modified substrates are cleaned using water droplets (gentle cleansing) to remove excess solution for uniform functional layer formation.

### Plasmonic vesicles preparation
Plasmonic vesicles were prepared via a two-step method following a previously reported method with minor modifications as follows[64]: first, Dipalmitoylphosphatidylcholine (DPPC) and cholesterol in a 4:1 molar ratio were dispersed in chloroform and dried with N$_2$, followed by overnight evaporation under a vacuum. The dry lipid film was then dispersed in 10 mM PBS containing calcein 75 mM for 1 h and subsequently extruded through 400 nm polycarbonate membranes for 11 passes using Avanti Mini Extruder (Avanti Polar Lipids). Free calcein was removed by centrifugation at 5000 g for 10 min and then washed with PBS three times. Second, gold nanoparticles were decorated onto DPPC liposome using the in-situ gold reduction method. Aqueous solutions of gold chloride (10 mM) and ascorbic acid (40 mM) were prepared. Gold chloride solution was added and gently mixed with liposome suspension (1.5 mM lipid concentration) in a molar ratio of 1:4 until uniformly distributed, followed by the addition of the same volume of ascorbic acid solution. Following reduction, plasmonic vesicles were separated from unreacted ascorbic acid and gold

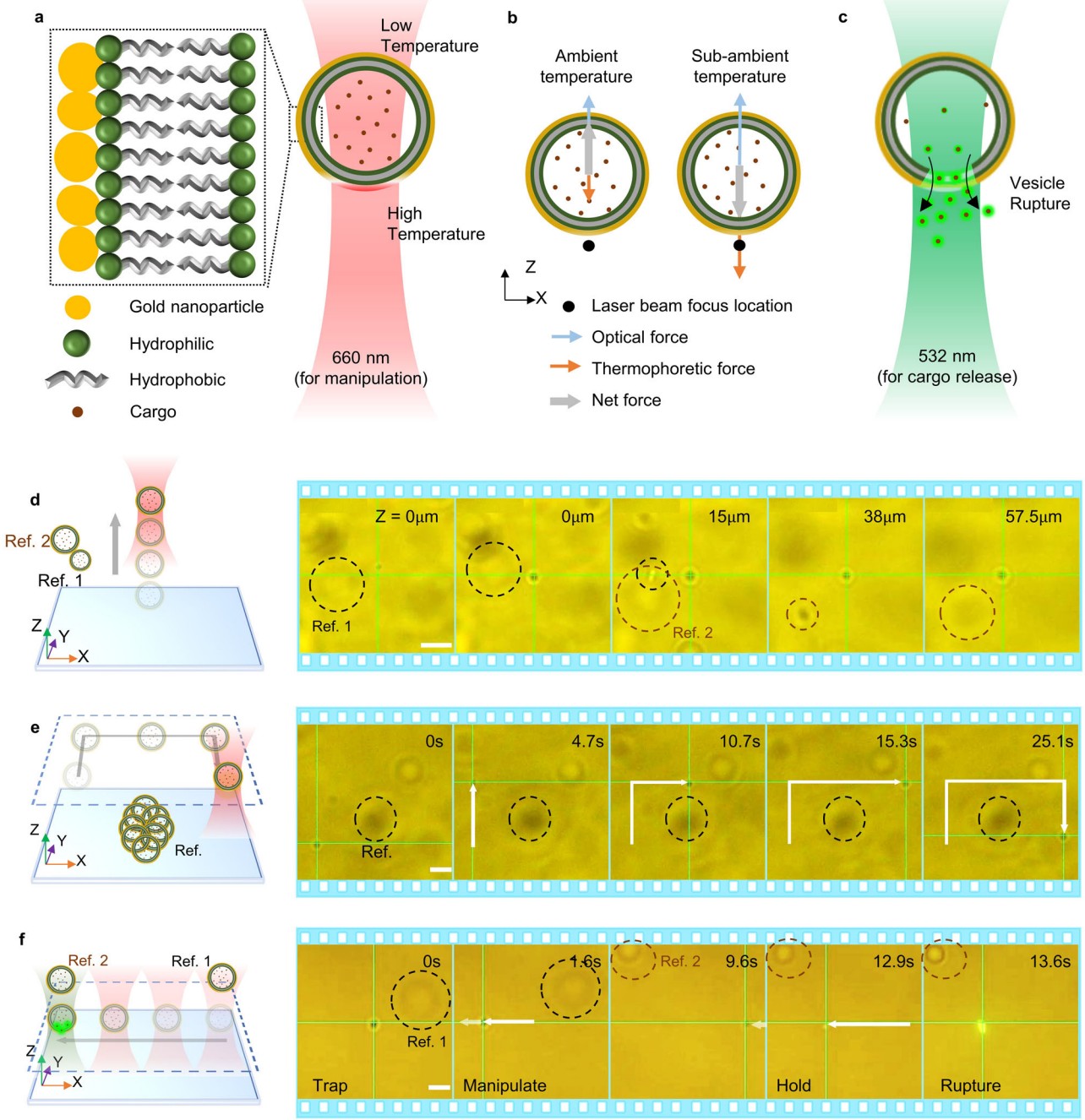

**Fig. 4 | 3D manipulation of plasmonic vesicles using HOTTs. (a)** Schematic of a plasmonic vesicle under 660 nm excitation. **b** Force analysis of the plasmonic vesicle at ambient temperature and sub-ambient temperature. **c** Schematic of vesicle rupture and cargo release due to 532 nm laser beam excitation (532 nm). Schematic and optical images showing **(d)** the levitation of a plasmonic vesicle, **(e)** in-plane manipulation of a plasmonic vesicle, and **(f)** manipulation of the vesicle and subsequent rupture for controlled cargo release. The power of the 660 nm laser for vesicle manipulation is 0.67 mW with a beam radius of 814 nm. The power of the 532 nm laser for vesicle rupture is 0.1 mW with a beam radius of 810 nm. Scale bars: 5 μm.

chloride by centrifugation (5000 g, 10 min) and then stored at 4 °C until use. For trapping experiments, plasmonic vesicles that are in the sub-micrometer regime (800–1500 nm) are used due to their easy visualization under the microscope.

## Optical setup

For substrate-based trapping, a 532 nm laser beam (Laser Quantum Ventus 532) is passed through a 5X beam expander and directed into the objective (Nikon Plan Fluor 40x, NA 0.75) of an optical microscope (Nikon Ti-E) through a series of reflective mirrors. The liquid sample containing the target objects are loaded into a 120 μm thick spacer that

acts as a microchamber. A charged coupled device (CCD – Nikon DS-Fi3) is used to visualize and record particle-trapping videos. Phase camera (Phasics, SID4 Bio) is used to evaluate the temperature increase because of laser heating of the substrate. The sample is placed on an aluminum sample holder that provides edge-support on all sides of the coverslip. The Peltier thermoelectric cooler (Laird Thermal Systems SH10-23-06-L1-W4.5) along with the aluminum heat sink is then rested on top of the glass coverslips. The weight of the heat sink ensures perfect contact of the thermoelectric cooler with the sample and the temperature resistance across the interface is assumed to be negligible. An annular cooler is selected to enable the light path through the

device. A corresponding through-hole is drilled into the heat sink, for white light to travel through the sample and reach the camera.

## Reporting summary

Further information on research design is available in the Nature Portfolio Reporting Summary linked to this article.

## Data availability

The data in Fig. 2a, b, e are available in the Source Data file. All microscopic images are taken from their corresponding raw data that are formatted and attached as Supplementary Data. All other data is available from the corresponding author upon request. The microscopic images provided in Fig. 2c, d are representative images during the estimation of trap ratio. For Fig. 2c, 49 and 123 different particles were studied at ambient temperature (27 °C) and sub-ambient temperature, respectively. For Fig. 2d, 39 and 37 different particles were studied at ambient temperature and sub-ambient temperature, respectively. The microscopic images provided in Supplementary Fig. 5 are representative of the shape and morphology of singular RBC's and are visually consistent across all experiments performed with RBCs. All microscope images shown in Figs. 3 and 4 and Supplementary Fig. 3 are representative of at least 5 distinct trials of their respective experiments. Source data are provided with this paper.

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

## Acknowledgements

The authors thank the Texas Advanced Computing Center (TACC) at the University of Texas at Austin for providing supercomputer resources used in the evaluation of optical forces in this work. The authors acknowledge the use of Autodesk 3dsMax 2022 for creating images Fig. 3a, d, f, and Supplementary Fig. 2a, b. The authors also acknowledge following funding support from NIH and NSF: National Institute of General Medical Sciences of the National Institutes of Health (R01GM146962, Y.Z.). National Science Foundation (ECCS-2001650, Y.Z.). National Institute of Neurological Disorders and Stroke of the National Institute of Health (RF1NS110499, Z.Q.), National Institute of General Medical Sciences (R35GM133653, Z.Q.). National Science Foundation (2123971, Z.Q.).

## Author contributions

P.S.K. and Y.Z. conceived the idea. P.S.K., performed the experiments and analyzed the data with assistance from Z.C., H.D., J.L., Y.K. and S.H. X.L. and Z.Q. synthesized plasmonic vesicles. Y.Z. supervised the project. P.S.K., J.L. and Y.Z. drafted the manuscript. All authors participated in the review and editing of the manuscript.

## Competing interests

The authors declare no competing interests.
