## [Peer Review File · Nature Communications]

Hypothermal opto-thermophoretic tweezersReviewer #1 (Remarks to the Author):

This paper study a novel design for thermophoretic accumulation of particles, where the trap is realized by cooling instead of the common laser heating. As a main result the authors find that the trapping efficiency increases at lower temperatures, thus confirming the well-known (but poorly understood) thermophilic behavior reported 20 years ago by Piazza. The authors propose various applications mainly for temperature sensitive systems.

This is a nice bit of work, based on an original idea, reporting novel results especially at temperatures approaching 4°C, and discussing possible applications.

Though I am not an expert in this field, I have the impression that the experiments and the data analysis are sound. My main criticism concerns the presentation and the discussion.

1) The setup is not quite clear. In Fig. 2a the red spot shows the heated area. According to the temperature map, in Fig. 2b the whole coverslip is at 4 °C? The spot where the laser hits the coverslip is even at a slightly higher temperature? Then I don't understand the direction of F_{th} , which should point towards the lowest T. Or is the arrow the superposition of thermophoresis and tweezer forces?

In any case, Fig. 2b and 2d should clearly show the temperature profile along the coverslip and perhaps give a hint whether and how T varies in vertical direction.

2) Why do the authors use NaOH solution? In previous papers a few years ago, they have shown that mobile anions such as OH may reverse the thermophoretic motion. I wonder whether this could interfere with the present setup, and render the interpretation of the data more intricate.

3) The discussion of the determination of the Soret coefficient in SM note 5 is a bit confusing. I don't quite understand the meaning of the thermophoretic force. I guess like in electrophoresis, "phoresis" means that there is no force on the particle. In my view, Eq. (S10) would make more sense when multiplied with the mobility D/kT , resulting in $u = -D S T \text{ grad}T$. Or do I miss something?

4) Perhaps Fig. S6d should be moved to the main text, I find this rather interesting. I understand that fundamental issues are not the main focus of this paper, still it would be nice to compare to Piazza's measurements [EJP B 2006].

5) About 10 years ago, the Cichos group designed kind of a thermal Paul trap where the particle moves in a parabolic T profile, and gets trapped at the coldest spot. The experiment is quite different from the present one, perhaps the authors wish to have a look, the paper is from 2013.

Reviewer #2 (Remarks to the Author):

In this work, Zheng and his co-authors harness the environmental temperature-dependent Soret coefficient for optothermal trapping. Although the temperature-dependency of Soret coefficient has been proposed and verified more than ten years ago, it is not well exploited in low-power and non-invasive optical trapping. Thus, it is overall an interesting work. The trapping of RBC and the 3D trapping of plasmonic vesicle is impressive. While the work may meet the publication standards of nature communications, there are still a couple of points which should be carefully addressed before it can be accepted for publication.

1. The authors describe the temperature-dependent Soret coefficient by the empirical equation, but did not provide any explanation why the temperature changes the sign of Soret coefficient. In addition, why the T^* varies for different materials? Are all particles have a negative Soret coefficient at a significantly low temperature above zero degree?

2. The influence of particle concentration on the trapping probability is interesting. Why the trapping probability is lowest at 2%? Detail discussion should be provided to understand the impact at different concentration. In term of particle-particle interaction at high concentration, it

occurs at all trapping experiments regardless of the working mechanism. For example, it should also take place in optical tweezers?

3. The authors show that the trapping phenomenon is different for RBC at different tonicity solution. I suppose that it stems from the different Soret coefficients. So, why the Soret coefficient changes with the tonicity of solution?

4. The 3D trapping of plasmonic vesicle is interesting. How about vesicle loaded with other cargoes, e.g., small molecules. I suppose that 3D trapping mechanism failed in such configuration. Any possibility to achieve 3D trapping of non-plasmonic targets?

5. Minor issues: scale bar is missing in some figure in the supporting information.

Reviewer #3 (Remarks to the Author):

In this manuscript, the authors presented a hypothermal optothermophoretic tweezers (HOTTs) to achieve low-power trapping of diverse colloids and biological cells in their native fluids. HOTTs exploit an environmental cooling strategy to simultaneously enhance the thermophoretic trapping force at sub-ambient temperatures and suppress the thermal damage to target objects. This HOTTs seems to be a typical optothermal tweezers with an environmental cooling. The addition of environmental cooling provides benefits of better trapping and less toxicity to biological samples. The results seem to be original.

1. A main concern of this HOTTs for biological application is the temperature. The HOTTs works great at low temperature of 4C while the local temperature near the light exposed spot is much higher. Such high temperature gradient is not good for biological samples or cells, their physiological function will be significantly affected by such high temperature gradience. In such low environmental temperature of 4C, cells will not function normally either, limiting the potential of this technology. In order to remove such toxicity concern, it would be important to prove the safety of the technique by conducting basic biological assays to test metabolic activity and reproducibility function after treatment.

It would be important to know the temperature at and near the light spot at different power level.

2. The authors claim that they can trap erythrocytes in different tonicities at extremely low laser power, while retaining their cellular integrity and biophysiochemical. They did demonstrate the trapping of the red blood cells without lysing them, but their biochemical functions are not characterized and it is unknown if they still retain normal functions. Intact cell membrane structure does not necessarily mean their normal biochemical function. It may be a good idea to conduct basic assays to test metabolic activity and reproducibility function after treatment.

3. It may be a good idea to add concentration label of x axis in fig. 2e.

Reviewer #1 (Remarks to the Author):

This paper study a novel design for thermophoretic accumulation of particles, where the trap is realized by cooling instead of the common laser heating. As a main result the authors find that the trapping efficiency increases at lower temperatures, thus confirming the well-known (but poorly understood) thermophilic behavior reported 20 years ago by Piazza. The authors propose various applications mainly for temperature sensitive systems. This is a nice bit of work, based on an original idea, reporting novel results especially at temperatures approaching 4°C, and discussing possible applications. Though I am not an expert in this field, I have the impression that the experiments and the data analysis are sound. My main criticism concerns the presentation and the discussion.

We thank the reviewer for the positive comments on our work. We have included the point-to-point responses to these remarks.

1. The setup is not quite clear. In Fig. 2a the red spot shows the heated area. According to the temperature map, in Fig. 2b the whole cover slip is at 4 °C? The spot where the laser hits the coverslip is even at a slightly higher temperature? Then I don't understand the direction of F_{th} , which should point towards the lowest T. Or is the arrow the superposition of thermophoresis and tweezer forces? In any case, Fig. 2b and 2d should clearly show the temperature profile along the coverslip and perhaps give a hint whether and how T varies in vertical direction.

We thank the reviewer for the comment. The arrows on the schematic indicate the net thermophoretic force on the particle. At both ambient and sub-ambient temperatures, the laser heating causes a local temperature hotspot at the center of the laser beam. At ambient temperature, the particles experience a thermophobic force on the particle due to a positive Soret coefficient of the particle, and F_{th} points towards the lower T (away from the laser beam). Upon reducing the environmental temperature using our custom setup, the thermal response of the particle changes from thermophobic to thermophilic, leading to a negative Soret coefficient and a F_{th} towards the higher T (towards the laser beam). In this case, the laser heating enables the particle to drift towards the hotspot, thereby trapping the particle at the hotspot. We indicated the environment temperature and the corresponding temperature increase from the environmental temperature on laser heating in our schematic. For more clarity, we took the suggestion from the reviewer and included a new SI figure detailing the simulated

temperature increase for a given laser power and at ambient ($27\text{ }^{\circ}\text{C}$) and sub-ambient ($4\text{ }^{\circ}\text{C}$) temperature.

Changes made: An additional SI figure is incorporated.

Figure S1: Temperature distribution due to laser heating thin AuNI on glass substrate: a) Perspective view of laser heating the sample at ambient temperature (left panel, $27\text{ }^{\circ}\text{C}$) and sub-ambient temperature (right panel, $4\text{ }^{\circ}\text{C}$). Color bar indicates the temperature in K. The dark gray and green rectangles indicate the substrate and perpendicular to the substrate respectively. b,c) Temperature distribution along the substrate (b) and vertical to the substrate (c) at ambient (i) and sub-ambient temperature (ii). The one-dimensional temperature profiles highlighted using lines on (i) and (ii) and indicated in (iii). Laser power: 0.45 mW . Scale bars: $5\text{ }\mu\text{m}$.

2. Why do the authors use NaOH solution? In previous papers a few years ago, they have shown that mobile anions such as OH may reverse the thermophoretic motion. I wonder whether this could interfere with the present setup, and render the interpretation of the data more intricate.

We thank the reviewer for the comment. PS particles in DI water do not exhibit trapping at ambient temperature (as shown in Figure 1) and range of environment temperatures for quantitative analysis was limited. Therefore, we used PS particles dispersed in $\text{NaCl}_x\text{OH}_{1-x}$ for quantitative analysis of the effect of environment temperature on the trapping stiffness. The use of Na^+ , Cl^- , and OH^- ions alters the thermophoretic response as shown in previous reports due to thermoelectric fields within the solution.¹ Despite the intricate nature of the force field, reducing the environment temperature enables trapping in many situations. To demonstrate the capability of our strategy, the trapping ability at reduced environment temperature is validated by a series of experiments to observe $\text{NaCl}_x\text{OH}_{1-x}$ for varying 'x' (between 0 and 1) and included it as an additional SI figure.

Changes made: *Main manuscript:* Line 115

Figure 2a shows the trajectories of a $1\ \mu\text{m}$ PS particle trapped in $3\text{mM NaCl}_{0.2}\text{OH}_{0.8}$ solution at varying temperatures. Here, we use an electrolyte solution dominated by NaOH to ensure trapping even at ambient temperature to determine the trend of the trapping strength over a higher range of temperatures. As the temperature reduces, the particle becomes more confined with respect to the laser beam center. Figure 2b further shows the calculated trapping stiffnesses of trapped particles at varying environment temperatures with different laser powers (0.05 mW, 0.14 mW, and 0.24 mW), sizes ($1\ \mu\text{m}$ and $9.5\ \mu\text{m}$), materials (PS and SiO_2), and solutions (DI water and electrolytes). In all the conditions, the trapping stiffness increases by 3-5 times with the reduced environmental temperature, showing the versatility of HOTTs. Moreover, by varying the composition of the electrolyte ('x' in $\text{NaCl}_x\text{OH}_{1-x}$), we demonstrate that trapping ability is induced in a wider range of compositions at reduced environmental temperature (Figure S4).

Changes made: *SI:* A new figure is added.

Figure S4: Trapping vs Repulsion of 1 μm PS particles in varying composition of 3 mM $\text{NaCl}_x\text{OH}_{1-x}$: As the temperature reduces, the trapping zone increases considerably. At ambient temperature (27 $^\circ\text{C}$), trapping is observed only in $\sim 20\%$ of the compositions ($x \leq 0.2$), however, reducing the ambient temperature to 4 $^\circ\text{C}$ increasing the trapping range to $\sim 80\%$ of the compositions ($x \leq 0.8$).

- The discussion of the determination of the Soret coefficient in SM note 5 is a bit confusing. I don't quite understand the meaning of the thermophoretic force. I guess like in electrophoresis, "phoresis" means that there is no force on the particle. In my view, Eq. (S10) would make more sense when multiplied with the mobility D/kT , resulting in $u = -D \nabla T$. Or do I miss something?

The authors would like to thank the reviewer for this comment. For this article, the term 'thermophoretic force' includes all the effects arising from temperature gradients, such as the inherent thermal drift of the particle, thermoelectric force, thermoosmotic force and so on. As the environment temperature changes, the net thermophoretic force (an intricate combination of all thermal gradient forces) depends on the temperature, altering its direction and magnitude of the force.

The net force acting on the particle is indeed zero as the Stokes drag force neutralizes other non-zero forces acting on the particle. For instance, a particle exhibiting electrophoresis experiences an electrostatic force, which is balanced by the Stokes drag force ($C \cdot 6\pi\mu r v$) also given in terms of the diffusion coefficient as $k_B T \cdot v/D$, where $D = C \cdot k_B T / 6\pi\mu r$, C being a correction factor. Here, the driving force is the electrostatic force given as qE , which is balanced out by the Stokes drag force, resulting in the evaluation of electrophoretic drift velocity.

Similarly, in our work, the term thermophoretic force indicates the net force due to thermal gradient acting on the particle. However, a minor contribution of the total force acting on the particle is also the optical force due to the laser beam incidence. The drag velocity visualized under the microscope gives us the net force on the particle, including the optical and thermophoretic force, the latter – a combination of all thermal gradient-induced forces. We understand that the optical force is negligible in our work due to the low laser power, but we would still like to employ force-related equation to create a more valid framework when extending this work to other particles such as metallic and light-absorbing particles which experience a significantly higher optical force at lower laser power (such as the plasmonic vesicles in this work and Si particles in ²). Eq (S10) would lead to a slightly inaccurate estimation of the Soret coefficient if the net drag velocity is equated only to the thermophoretic force.

Changes made: *Main manuscript*: Line 74

“Here, thermophoretic force includes the contributions from all thermal-gradient-induced forces such as thermo-electricity, thermo-osmosis, and thermo-diffusion.” is added.

Changes made: *SI*: Line 167

“Please note that the u_{th} is different from u , the former being the thermophoretic drift velocity and the latter being the escape velocity ($u_{th} = u$ when optical force = 0). Here, we use a force-based approach to include the non-zero contribution of the optical force, which may be significant, while extending this work to other metallic and light-absorbing particles.” is added.

4. Perhaps Fig. S6d should be moved to the main text, I find this rather interesting. I understand that fundamental issues are not the main focus of this paper, still it would be nice to compare to Piazza's measurements [EJP B 2006].

We thank the reviewer for the suggestion to compare it to Piazza's measurements. We included the Soret coefficient data from literature as a supplementary table. We respectfully disagree with the reviewer on including this result in the main text, mainly because this result can be possible only for particles that are thermophoretically attracted by the laser beam even at room temperature. Since PS particles are exhibiting thermophobic behavior in DI water at room temperature, we utilized SiO₂ particles that demonstrate thermophilic behavior up to 32 °C in DI water. The suggested reference and other related references mainly evaluate nanometer-scale entities and have Soret coefficients on a

different order ($0.01 - 100 \text{ K}^{-1}$), while our measured Soret coefficient is of the order of 1000 K^{-1} mainly due to micrometer size of the particle. Moreover, the method used in Piazza's measurements are one-dimensional temperature gradient spanning much higher length scale compared to the particle size. We clarified our statements in the main manuscript.

Changes made: *Main manuscript*: Line 126

“To quantitatively evaluate the effect of environmental temperature on the thermophoretic behavior, we measured the drag velocity of trapped **SiO₂ particles in DI water** to extract the Soret coefficient (S_T) at different temperatures (Supplementary Note 4). **The experimental data fit nicely with the empirical formula (Equation 2). It is noted that the Soret coefficient of the particle is tuned from $-0.1 \times 10^3 \text{ K}^{-1}$ to $-2 \times 10^3 \text{ K}^{-1}$ after reducing the temperature from $31 \text{ }^\circ\text{C}$ to $4 \text{ }^\circ\text{C}$, which is of the same order of magnitude for PS particles of similar size.**”³

5. About 10 years ago, the Cichos group designed kind of a thermal Paul trap where the particle moves in a parabolic T profile, and gets trapped at the coldest spot. The experiment is quite different from the present one, perhaps the authors wish to have a look, the paper is from 2013.

We thank the reviewer for highlighting the reference from Prof. Cichos's group. We are assuming that the reviewers are referring to (Braun, M.; Cichos, F. Optically Controlled Thermophoretic Trapping of Single Nano-Objects. ACS Nano 2013, 7, 11200-11208.). The paper enables the trapping of thermophobic objects in the colder region as the heating spot moves in a circle. Our work enables the trapping of different objects by inducing thermophilicity in diverse solutions through environmental temperature control, while Prof. Cichos' work enables the confinement of thermophobic particles. Moreover, these traps do not possess the ability to manipulate, mainly because of the static plasmonic structures that are required to generate the temperature hotspots. We included the suggested reference and other related references in the introduction.

Changes made: *Main manuscript*: Line 57

“**Without any solution modification, specialized traps such as thermal Paul trap and anti-Brownian electrokinetic trap were implemented to trap thermophobic objects with limited accuracy and manipulation abilities.**⁴⁻⁶” is added.

Reviewer #2 (Remarks to the Author):

In this work, Zheng and his co-authors harness the environmental temperature-dependent Soret coefficient for optothermal trapping. Although the temperature-dependency of Soret coefficient has been proposed and verified more than ten years ago, it is not well exploited in low-power and non-invasive optical trapping. Thus, it is overall an interesting work. The trapping of RBC and the 3D trapping of plasmonic vesicle is impressive. While the work may meet the publication standards of nature communications, there are still a couple of points which should be carefully addressed before it can be accepted for publication.

We thank the reviewer for highlighting the importance of our work. We are attaching our responses to the reviewer's comments in detail.

1. The authors describe the temperature-dependent Soret coefficient by the empirical equation, but did not provide any explanation why the temperature changes the sign of Soret coefficient. In addition, why the T^* varies for different materials? Are all particles have a negative Soret coefficient at a significantly low temperature above zero degree?

We thank the reviewer for highlighting this missing point in our manuscript. The temperature-dependence of Soret coefficient used in this work is a well-established empirical concept, and many particles demonstrate thermophilicity at reduced temperatures. Along with the experiments in this work that show the enhanced thermophilic nature of different particles and biological objects in varying solution compositions, we are also highlighting Soret coefficients of different particles in diverse solutions from literature that show thermophilicity at lower temperatures as a table in supplementary information.

The temperature strongly affects the particle-solvent interactions. Particularly, the aqueous solvents used for most biological applications have dynamic hydrogen bond network into which colloids or cells are introduced. At the particle surface, the hydrogen bond network is reorganized to accommodate the particle, which depends strongly on the material.⁷ When a temperature gradient is applied, water molecules are thermo-polarized, resulting in an electric field. This is observed both in pure DI water as well as in electrolyte solutions.⁸⁻¹⁰ The temperature affects the directionality of the thermo-polarization, and the inversion of the Soret coefficient is mainly observed due to the quadrupolar component of the electric field, which also affects the density of water.⁸ Since water has

the highest density around 4 °C, we expect a negative Soret coefficient for most of the particles in DI water.

However, the dependence of the quadrupolar component on temperature is less dominant in electrolyte solutions due to other competing factors such as thermoelectricity and thermo-osmosis. Therefore, the negative coefficient might not always be possible in electrolyte solutions as shown in our new SI information (also added as a response to R1.2).

Changes made: *Main manuscript:* Line 79

“Mainly, temperature plays a significant role in affecting the particle-solvent interactions. The aqueous solvent used in several biological applications exhibits thermo-polarization of water molecules, creating a thermo-electric field that inverts as the temperature changes. This is mainly due to the strong dependence of the quadrupolar component of the thermo-electric field on the temperature, which is determined by the thermal expansion of water.⁸ The resultant S_T typically reduces with the decreasing temperature in the majority cases, which can be described by an empirical” is added.

Line 87:

“ T^* changes mainly with the particle material that affects the particle-solvent interactions. This is due to the reorganization of the hydrogen bond network of the solvent while accommodating the dispersed particle.⁷” is added.

Changes made: *SI:* The following table is added.

Supplementary Table S1: Literature survey of aqueous solutions of diverse particles exhibiting temperature-dependent Soret coefficient and following the empirical law $\left(S_T = S_{T,\infty} \left(1 - e^{-\frac{T^*-T}{T_0}} \right) \right)$

Sl.No	Particle/Ion/molecule	Size/concentration	Solvent	Temperature	Soret coefficient (K ⁻¹)	Reference	Follows empirical law?
1	Pullulan	5g/mL	Water	20	-0.06	11	Yes
				50	0.02		
2	PS	90 nm	5 mM NaCl in water	27	2	12	Yes
			5 mM NaOH in water	41-57	-1.25		No
3	DNA	5-50 bases	Water	5	-0.075 – -0.025	13	Yes
				70	0-0.05		
4	CoFe ₂ O ₃ core, γ -	13.6 nm	10 mM HNO ₃	20	-3.25	14	Yes

	Fe ₂ O ₃ shell			55	-0.5		
5	KI	1 mol/kg	Water	15	-0.001	15	Yes
				45	+0.001		
	NaI			15	-0.001		
				45	+0.0015		
	LiI			15	-0.004		
45	-0.002						
6	PS	2.5 μm	Water	10	-80	3	Yes
				37	250		
	Melamine	1.35 μm	Water	10	-90		
				57	15		
7	PS	30 nm	4 mM NaCl	5	-0.02	3	Yes
				40	0.24		
8	Streptavidin	39 g/L	Water	10	-0.028	16	Yes
				50	0.02		
	Biotin	0.94 g/L	Water	10	0.005		
				50	0.01		
9	Gold NPs	28 nm	Water	28	-0.01	17	Yes
				40	0.02		
10	PS-COOH	26 nm	Water (pH 6.5)	10	-0.35	18	Yes
				60	0.3		
11	WT lysozyme	-	Water	10	-0.01	18	Yes
				50	0.01		
12	PS-COOH	123 nm	1mM Tris-HCl buffer	10	-0.6	19	Yes
				45	1.5		
13	Lysozyme	7 g/l	400 mM NaCl	5	-0.02	20	Yes
				35	0.008		
14	Vesicles – DOPC, DPPC (and others)	1 μm	Water	5	-0.05	21	Yes
				55	0.45		

2. The influence of particle concentration on the trapping probability is interesting. Why the trapping probability is lowest at 2%? Detail discussion should be provided to understand the impact at different concentration. In terms of particle-particle interaction at high concentration, it occurs at all trapping experiments regardless of the working mechanism. For example, it should also take place in optical tweezers?

We thank the reviewer for the suggestion. Previously, it has already been determined that collective effects alter the thermophoretic response of the particles significantly compared to other responses such as electrophoresis, where the transport of colloidal particles is driven mainly by short-ranged flows.²² The work also clearly highlights how particles experiencing a unidirectional thermal gradient show a concentration-dependent thermophoretic drift. In our case, this concentration dependence is complicated because of the laser-induced radial thermal gradient spanning only a few times the particle size. At our reported concentrations, the particle in the trap is surrounded by several particles, each of which is experiencing a unique thermal gradient (and hence a unique thermal drift) which

eventually alters the thermophoretic response on the particle. The spatial variance in the temperature gradient can induce non-monotonous trapping efficiency due to a non-monotonous Soret effect, which was also observed in unidirectional temperature gradients (Figure 1 in ²³).

The particle-particle interactions at high concentrations indeed affect the trapping ability of optical tweezers. Optical tweezers in turbid media is highly challenging where the environment is constantly moving, however, wavefront correction method has been developed to overcome the limitation to navigate the particle out of the turbid media,²⁴ which requires a feedback loop and a probe particle to enable the phase correction. In our work, we are demonstrating the ability of trapping within the turbid media at high concentrations, especially when the thermophoresis has a non-linear dependence on concentration.

Changes made: *Main manuscript*: Line 136:

“However, the non-linear nature of the thermophoretic response of the particles due to collective effects and spatially varying temperature gradient makes it challenging to achieve trapping at all concentrations.” is added.

Line 143:

“At high colloid concentrations, the thermoelectric field due to the charged particles is locally altered by the inter-particle interactions. In one-dimensional temperature gradients generated by a typical thermophoresis setup, the colloid’s concentration and zeta potential are sufficient to determine the thermophoretic response of the particle.²² However, in temperature gradients spatially varying over the same length scale as the particle’s size ($\sim 1\mu\text{m}$), the thermoelectric response is additionally dependent on the surrounding particle’s position distribution, which continuously varies due to the Brownian motion of the particle, thus requiring statistical analysis to determine the thermoelectric response of single particles. Accordingly, we focus the laser beam on single particles to determine the instantaneous trapping probability, which fluctuates between 10-40 % at the ambient temperature at concentrations from $\Phi > 0.01$, (Φ is the volume fraction of colloids, Figure 2e). As the temperature decreases to 4 °C, the enhanced thermophilic nature of the particles increases the trapping probability to 100 %. At low colloid concentration ($\Phi < 0.01$) in the single-particle limit, the PS particle undergoes a transition from repelling at ambient temperature to trapping at low temperatures (Figure 2d). Although single-particle trapping (or repulsion) is achieved without other particles in the trapping zone, the collective effects due to inter-particle interactions still exist at

volume fractions as low as ~ 0.001 .²² At lower concentrations, a delicate balance between the collective effects and individual particle's response occurs, and an inflection point is typically observed with changing colloidal concentration,²³ resulting in the least trapping performance at 2 % at ambient temperature. However, when the temperature is reduced to 4 °C, ...” is added.

3. The authors show that the trapping phenomenon is different for RBC at different tonicity solution. I suppose that it stems from the different Soret coefficients. So, why the Soret coefficient changes with the tonicity of solution?

We thank the reviewer for this question. Soret coefficient (and the thermal diffusion coefficient) are strong functions of several parameters such as ion concentrations, ion type, temperature, dissociation constants, zeta potential and so on.²⁵ As the tonicity of the solution changes, two major phenomena occurs. First, the ionic concentrations changes which alters the thermoelectric field and thermo-osmotic flow around the cells, which changes the thermophoretic response of the cell, similar to other micro and nanoparticles.²⁶ Second, the shape and size of the cell changes, which again changes the thermophoretic response significantly.²⁷ Additionally, the red blood cells continuously exchange water with its surroundings, which might result in unknown thermophoretic phenomena due to thermal regulation of the cell with its surroundings.²⁸ As the ion concentration changes, the effective interplay between the thermoelectricity and thermo-osmosis changes with temperature, altering the net interaction potential between the cell and its surroundings, ultimately changing the Soret coefficient of the cell.

Changes made: *Main manuscript*: Line 207

“Altering the tonicity of the extracellular fluid changes the thermoelectric and thermo-osmotic fields, as well as the effective shape and size of the cells, which changes the Soret coefficient of the cells.^{26,28} Despite the variation, erythrocytes are trapped by HOTTs in all tonicities, while maintaining their structural integrity.” is added.

4. The 3D trapping of plasmonic vesicle is interesting. How about vesicle loaded with other cargoes, e.g., small molecules. I suppose that 3D trapping mechanism failed in such configuration. Any possibility to achieve 3D trapping of non-plasmonic targets?

We thank the reviewer for their comment. The gold nanoaggregates on the outer portion of the vesicle enable the light-to-heat conversion and create a localized temperature gradient around the vesicle. The vesicle surface interacts with its environment in this spatially varying temperature

gradient and the cargo inside the vesicle is not expected to alter the vesicle's interaction potential with its surroundings. Therefore, the delivery of cargoes using plasmonic vesicles will remain effective regardless of the cargo material.

The reviewer raises a good question on the 3D trapping of non-plasmonic targets. 3D optothermal manipulation of non-plasmonic targets' is not possible since there is no heating element involved (except when the targets are lossy materials such as silicon). Based on an earlier work, regular vesicles with no gold coating can be manipulated in three-dimensions using a light-absorbing particle as a delivery vehicle.

Changes made: *Main manuscript*: Line 260

“The manipulation and rupture of vesicles depend on the interfacial properties between the gold nanoaggregates and solvent, which is independent of the cargoes within the vesicles. Therefore, HOTTs can be extended to deliver diverse cargoes using plasmonic vesicles as the delivery agent.” is added.

Changes made: *Main manuscript*: Line 283

“HOTTs can additionally be extended to non-plasmonic targets by using a plasmonic or a light-absorbing particle as a delivery agent.²” is added.

During the revision, we also noticed a minor error in the schematic of the vesicle, where we incorrectly represented the vesicle membrane using inter-digitated lipids. We corrected the schematic in our updated manuscript, where the lipids forming the membrane are represented as a bilayer.

Changes made: *Main manuscript*:

Figure 4a is changed from:

to:

5. Minor issues: scale bar is missing in some figure in the supporting information.

We thank the reviewer for highlighting the missing information. All figures have been updated with scale bars.

Reviewer #3 (Remarks to the Author):

In this manuscript, the authors presented a hypothermal optothermophoretic tweezers (HOTTs) to achieve low-power trapping of diverse colloids and biological cells in their native fluids. HOTTs exploit an environmental cooling strategy to simultaneously enhance the thermophoretic trapping force at sub-ambient temperatures and suppress the thermal damage to target objects. This HOTTs seems to be a typical optothermal tweezers with an environmental cooling. The addition of

environmental cooling provides benefits of better trapping and less toxicity to biological samples. The results seem to be original.

We thank the reviewer for the positive comments on our work. We have included the point-to-point responses to these remarks.

1. A main concern of this HOTTs for biological application is the temperature. The HOTTs works great at low temperature of 4C while the local temperature near the light exposed spot is much higher. Such high temperature gradient is not good for biological samples or cells, their physiological function will be significantly affected by such high temperature gradience. In such low environmental temperature of 4C, cells will not function normally either, limiting the potential of this technology. In order to remove such toxicity concern, it would be important to prove the safety of the technique by conducting basic biological assays to test metabolic activity and reproducibility function after treatment. It would be important to know the temperature at and near the light spot at different power level.

We thank the reviewer for this comment. We agree that high temperature gradients and cold temperatures may not be ideal for biological applications. In our attempt to comprehend the impact of high temperature gradients, we opted to employ red blood cells instead of yeast cells that are known to be more resistant and stable under varying environmental conditions, and employed the red blood cell membrane's structural integrity as a metric to determine their stability. We adopted immunofluorescence protocols to identify the GLUT-1 protein, which is prominently expressed on the membranes of viable red blood cells. We used a single step staining process to label the GLUT-1 proteins with FITC-conjugated antibody (GLUT-1-FITC, Fabgennix) using their protocols as a starting step. Although we endeavored to proceed in this direction, we encountered significant challenges in optimizing the assay protocols that are compatible with our samples used for laser incidence.

Mainly, the difficulty was in identifying the cells exposed to the laser beam. In our earlier experiments, we coated our thermoplasmonic substrates with albumin bovine serum (BSA) to prevent cell adhesion to the glass substrate and demonstrate the trapping ability. However, this coating prevented us from identifying the cells that were exposed to the laser beam during the immunofluorescence protocol. To overcome this obstacle, we removed the BSA coating and allowed the red blood cells to adhere to the glass coverslips. After repeated washing steps, we were able to

identify the cells that were exposed to the laser beam. Although we were unable to demonstrate the attraction of red blood cells towards the laser beam, the identification of the cells was easily achieved.

We then conducted several control experiments to optimize the incubation times and antibody concentration. Despite our efforts, we did not observe any fluorescence signals (even in control samples without any laser incidence). Further optimization is necessary for identification of metabolic activity using our protocols for single-cell assays.

We reached out to the company 'Human Bio Cells' that provided us with red blood cells for assistance in this study. During our correspondence, they suggested to us that red blood cells are inherently fragile and any significant or irreversible changes to their physiological conditions can cause the cell membrane to rupture. Our preliminary experiments, which were included in a previous submission, supported this observation.

We would like to suggest alternative strategies to minimize the impact of high temperature gradients on the physiological conditions of cells. Although high temperature gradients are necessary to exert strong trapping forces on cells, reducing the duration of laser exposure can help maintain the integrity of cells significantly.¹ Additionally, introducing a feedback loop and an automated beam expander can attract cells using higher temperature gradients while gradually increasing the beam waist at the focus as the cell approaches the laser beam center. This approach reduces the effect of higher temperature gradients and could be an effective way to minimize damage to cells during trapping.

Changes made: *Main manuscript*: Line 211

“While their structural integrity remains intact, further assay experiments are necessary to verify the preservation of cellular physiological conditions following laser exposure. Decreasing the duration of laser exposure would aid in maintaining the physiological conditions of the cells. Incorporating alternative engineering methods that employ focused laser beams to attract cells while gradually increasing the focal beam waist as cell approaches the laser beam (thus reducing the temperature gradient) would additionally contribute to the preservation of cellular integrity.” is added.

2. The authors claim that they can trap erythrocytes in different tonicities at extremely low laser power, while retaining their cellular integrity and biophysiochemical. They did demonstrate the trapping of the red blood cells without lysing them, but their biochemical functions are not characterized and it is unknown if they still retain normal functions. Intact cell membrane structure does not necessarily mean their normal biochemical function. It may be a good idea to conduct basic assays to test metabolic activity and reproducibility function after treatment.

We thank the reviewer for this suggestion and we agree that the comment on biophysiochemical interactions is not completely valid. We highlighted the challenges we faced during single-cell characterization regarding their metabolic functions. Therefore, we changed the relative portions, in the manuscript.

Changes made: *Main manuscript*: Line 188

We removed the claim of bio-physiochemical functions from our manuscript.

“Here, we demonstrate the capability of HOTTs for biological applications by trapping erythrocytes in different tonicities at extremely low laser power, while retaining their ~~cellular-structural~~ integrity ~~and bio-physiochemical functions~~”.

Line 279:

“~~Although we successfully demonstrate the trapping of red blood cells in diverse tonicities while retaining the structural integrity, additional studies are essential to successfully characterize the metabolic activity of singular cells after laser exposure.~~” is added to explain the readers clearly about our current limitation.

3. It may be a good idea to add concentration label of x axis in fig. 2e.

We thank the reviewer for the suggestion. We added the concentration label on the X-axis.

References:

- 1 Deng, J. L., Wei, Q., Zhang, M. H., Wang, Y. Z. & Li, Y. Q. Study of the effect of alcohol on single human red blood cells using near-infrared laser tweezers Raman spectroscopy. *Journal of Raman Spectroscopy* **36**, 257-261, doi:10.1002/jrs.1301 (2005).
- 2 Lin, L. *et al.* Opto-thermoelectric pulling of light-absorbing particles. *Light: Science & Applications* **9**, doi:10.1038/s41377-020-0271-6 (2020).
- 3 Helden, L., Eichhorn, R. & Bechinger, C. Direct measurement of thermophoretic forces. *Soft Matter* **11**, 2379-2386, doi:10.1039/c4sm02833c (2015).
- 4 Braun, M., Würger, A. & Cichos, F. Trapping of single nano-objects in dynamic temperature fields. *Phys. Chem. Chem. Phys.* **16**, 15207-15213, doi:10.1039/c4cp01560f (2014).
- 5 Wang, Q., Goldsmith, R. H., Jiang, Y., Bockenbauer, S. D. & Moerner, W. E. Probing Single Biomolecules in Solution Using the Anti-Brownian Electrokinetic (ABEL) Trap. *Accounts of Chemical Research* **45**, 1955-1964, doi:10.1021/ar200304t (2012).
- 6 Braun, M. & Cichos, F. Optically Controlled Thermophoretic Trapping of Single Nano-Objects. *ACS Nano* **7**, 11200-11208, doi:10.1021/nn404980k (2013).
- 7 Niether, D. & Wiegand, S. Thermophoresis of biological and biocompatible compounds in aqueous solution. *Journal of Physics: Condensed Matter* **31**, doi:10.1088/1361-648x/ab421 (2019).
- 8 Armstrong, J. & Bresme, F. Temperature inversion of the thermal polarization of water. *Physical Review E* **92**, doi:10.1103/physreve.92.060103 (2015).
- 9 Di Lecce, S., Albrecht, T. & Bresme, F. The role of ion–water interactions in determining the Soret coefficient of LiCl aqueous solutions. *Physical Chemistry Chemical Physics* **19**, 9575-9583, doi:10.1039/c7cp01241a (2017).
- 10 Ding, H., Kollipara, P. S., Lin, L. & Zheng, Y. Atomistic modeling and rational design of optothermal tweezers for targeted applications. *Nano Research* **14**, 295-303, doi:10.1007/s12274-020-3087-z (2021).
- 11 Kishikawa, Y., Wiegand, S. & Kita, R. Temperature Dependence of Soret Coefficient in Aqueous and Nonaqueous Solutions of Pullulan. *Biomacromolecules* **11**, 740-747, doi:10.1021/bm9013149 (2010).
- 12 Eslahian, K. A., Majee, A., Maskos, M. & Würger, A. Specific salt effects on thermophoresis of charged colloids. *Soft Matter* **10**, 1931, doi:10.1039/c3sm52779d (2014).
- 13 Reichl, M., Herzog, M., Götz, A. & Braun, D. Why Charged Molecules Move Across a Temperature Gradient: The Role of Electric Fields. *Physical Review Letters* **112**, doi:10.1103/physrevlett.112.198101 (2014).
- 14 Sehnem, A. L. *et al.* Temperature dependence of the Soret coefficient of ionic colloids. *Physical Review E* **92**, doi:10.1103/physreve.92.042311 (2015).
- 15 Mohanakumar, S., Kriegs, H., Briels, W. J. & Wiegand, S. Overlapping hydration shells in salt solutions causing non-monotonic Soret coefficients with varying concentration. *Physical Chemistry Chemical Physics* **24**, 27380-27387, doi:10.1039/d2cp04089a (2022).
- 16 Niether, D. *et al.* Thermophoresis: The Case of Streptavidin and Biotin. *Polymers* **12**, 376, doi:10.3390/polym12020376 (2020).
- 17 Shakib, S. *et al.* Microscale Thermophoresis in Liquids Induced by Plasmonic Heating and Characterized by Phase and Fluorescence Microscopies. *The Journal of Physical Chemistry C* **125**, 21533-21542, doi:10.1021/acs.jpcc.1c06299 (2021).
- 18 Putnam, S. A., Cahill, D. G. & Wong, G. C. L. Temperature Dependence of Thermodiffusion in Aqueous Suspensions of Charged Nanoparticles. *Langmuir* **23**, 9221-9228, doi:10.1021/la700489e (2007).

- 19 Braibanti, M., Vigolo, D. & Piazza, R. Does Thermophoretic Mobility Depend on Particle Size? *Physical Review Letters* **100**, doi:10.1103/physrevlett.100.108303 (2008).
- 20 Piazza, R., Iacopini, S. & Triulzi, B. Thermophoresis as a probe of particle–solvent interactions: The case of protein solutions. *Phys. Chem. Chem. Phys.* **6**, 1616-1622, doi:10.1039/b312856c (2004).
- 21 Talbot, E. L., Kotar, J., Parolini, L., Di Michele, L. & Cicuta, P. Thermophoretic migration of vesicles depends on mean temperature and head group chemistry. *Nature Communications* **8**, 15351, doi:10.1038/ncomms15351 (2017).
- 22 Majee, A. & Würger, A. Collective thermoelectrophoresis of charged colloids. *Physical Review E* **83**, doi:10.1103/physreve.83.061403 (2011).
- 23 Dhont, J. K. G. Thermodiffusion of interacting colloids. II. A microscopic approach. *The Journal of Chemical Physics* **120**, 1642-1653, doi:10.1063/1.1633547 (2004).
- 24 Čižmár, T., Mazilu, M. & Dholakia, K. In situ wavefront correction and its application to micromanipulation. *Nature Photonics* **4**, 388-394, doi:10.1038/nphoton.2010.85 (2010).
- 25 Kollipara, P. S., Chen, Z. & Zheng, Y. Optical Manipulation Heats up: Present and Future of Optothermal Manipulation. *ACS Nano* **17**, 7051-7063, doi:10.1021/acsnano.3c00536 (2023).
- 26 Würger. Thermal non-equilibrium transport in colloids. *Reports on Progress in Physics* **73**, doi:10.1088/0034-4885/73 (2010).
- 27 Zheng, F. Thermophoresis of spherical and non-spherical particles: a review of theories and experiments. *Advances in Colloid and Interface Science* **97**, 255-278, doi:[https://doi.org/10.1016/S0001-8686\(01\)00067-7](https://doi.org/10.1016/S0001-8686(01)00067-7) (2002).
- 28 Ferdinandus *et al.* Modulation of Local Cellular Activities using a Photothermal Dye-Based Subcellular-Sized Heat Spot. *ACS Nano* **16**, 9004-9018, doi:10.1021/acsnano.2c00285 (2022).

Reviewer #1 (Remarks to the Author):

The authors have clarified several points and provided a lot of additional information in ms and SI.

I recommend publication in Nature Comm.

Reviewer #2 (Remarks to the Author):

The authors have addressed my concerns and the revisions seem acceptable to me. Now I can recommend the publication of this work.

Reviewer #3 (Remarks to the Author):

Looks good to me.